# Assessing the Drivers behind the Structure and Diversity of Fish Assemblages Associated with Rocky Shores in the Galapagos Archipelago

Stijn Bruneel [1,2,*], Wout Van Echelpoel [1], Long Ho [1], Heleen Raat [1], Amber Schoeters [1], Niels De Troyer [1], Ratha Sor [1,3], José Ponton-Cevallos [1,4,5], Ruth Vandeputte [1], Christine Van der heyden [6], Nancy De Saeyer [1], Marie Anne Eurie Forio [1], Rafael Bermudez [5,7], Luis Dominguez-Granda [8], Stijn Luca [9], Tom Moens [2] and Peter Goethals [1]

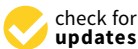



1  Department of Animal Sciences and Aquatic Ecology, Ghent University, Coupure Links 653, 9000 Ghent, Belgium; Wout.VanEchelpoel@UGent.be (W.V.E.); Long.tuanho@UGent.be (L.H.); heleen_raat@hotmail.com (H.R.); amber.schoeters@hotmail.com (A.S.); niels.detroyer@ugent.be (N.D.T.); sorsim.ratha@gmail.com (R.S.); josefernando.pontoncevallos@ugent.be (J.P.-C.); ruthvandeputte@hotmail.com (R.V.); nancy.desaeyer@ugent.be (N.D.S.); marie.forio@ugent.be (M.A.E.F.); peter.goethals@ugent.be (P.G.)
2  Marine Biology Research Group, Ghent University, Krijgslaan 281, 9000 Ghent, Belgium; tom.moens@ugent.be
3  Dean of Graduate School, Chea Sim University of Kamchaymear, No. 157, Preah Norodom Blvd, Phnom Penh 12156, Cambodia
4  Facultad de Ciencias de la Vida, Escuela Superior Politécnica del Litoral (ESPOL), Campus Gustavo Galindo, Guayaquil 09-01-5863, Ecuador
5  Galapagos Marine Research and Exploration, GMaRE. Joint ESPOL-CDF program, Charles Darwin Research Station, Galapagos Islands 200102, Ecuador; jrbermud@espol.edu.ec
6  Department of Biosciences and Industrial Technology, Research Centre of Health and Water Technology, HOGENT, University of Applied Sciences and Arts, Valentin Vaerwyckweg 1, 9000 Ghent, Belgium; christine.vanderheyden@hogent.be
7  Facultad de Ingeniería Marítima y Ciencias del Mar, Escuela Superior Politécnica del Litoral (ESPOL), Campus Gustavo Galindo, Guayaquil 09-01-5863, Ecuador
8  Centro del Agua y Desarrollo Sustentable, Facultad de Ciencias Naturales y Matemáticas, Escuela Superior Politécnica del Litoral (ESPOL), Campus Gustavo Galindo, Guayaquil 09-01-5863, Ecuador; ldomingu@espol.edu.ec
9  Department of Data Analysis and Mathematical Modelling, Ghent University, Coupure Links 653, 9000 Ghent, Belgium; stijn.luca@ugent.be
*  Correspondence: stijn.bruneel@ugent.be

**Abstract:** Oceanic islands harbor unique yet fragile marine ecosystems that require evidence-based environmental management. Among these islands, the Galapagos archipelago is well known for its fish diversity, but the factors that structure communities within and between its islands remain poorly understood. In this study, water quality, physical habitats and geographical distance were assessed as potential predictors for the diversity and structure of fish assemblages. Differences in the structure of fish assemblages of the two studied islands (Santa Cruz and Floreana) were most likely driven by temperature and nutrient concentrations. In the relatively highly populated island Santa Cruz, the structure of fish assemblages was more affected by water conditions than physical habitats while the contrary was true for the more pristine area of Floreana. A wide variety of species with different geographical origins were distributed over the different islands, which indicates that most fish species are able to reach the islands of the archipelago. However, temperature gradients and elevated nutrient levels cause large differences in the structure of local fish assemblages. In addition, in Santa Cruz nutrient concentrations were negatively correlated with $\alpha$ diversity. Since pollution is a clear pressure on the fish assemblages of oceanic islands, environmental management of the coastal areas is of paramount importance.

**Keywords:** coastal ecosystem; video monitoring; Tropical Eastern Pacific fish assemblage; Galapagos; water quality; anthropogenic pressure

## 1. Introduction

Due to their isolated position, oceanic islands harbor some of the last near-pristine marine ecosystems and associated fish assemblages [1,2]. Anthropogenic pressures, such as fishery, pollution, and the introduction of invasive species, threaten, however, many of these already fragile marine communities, and sound evidence-based management is required to protect them from extinction [3,4]. Among these oceanic islands, the Galapagos archipelago is key due to its exceptionally rich biodiversity and its role as stepping stone between the Tropical Eastern and Central Pacific [5,6]. Although the rich biodiversity of the Galapagos archipelago has been attributed to it being located at the intersection of multiple warm and cold ocean currents [7], it remains unclear what the drivers behind the diversity and structure of the local fish assemblages are.

The observed differences in the marine communities seem to coincide with sharp differences in environmental conditions, which have given rise to the delineation of multiple bio-geographical regions [8,9]. The strong regional divisions in fish assemblages observed by Edgar et al. (2004) were considered to reflect both the local environmental conditions and connectivity of fish larvae with external source regions, such as the Indo-Pacific, Panamic and Peruvian region [5]. However, at the same time, the high species richness on the far-northern isolated islands of Wolf and Darwin suggested high immigration rates and a strong connectivity of fish assemblages between the islands [5]. In the case of high inter-island connectivity, it is expected that fish are able to reach the different available habitats and that their prevalence is mainly determined by local environmental conditions, rather than by dispersal limitation. Based on differences in temperature, Harris (1969) identified five potential bio-geographical regions in the archipelago [8], but these were significantly different from the regions proposed by Edgar et al. (2004), which were based on differences in the structure of fish and macroinvertebrate assemblages [5]. According to Edgar et al. (2004), there was no sound evidence to subdivide the area east of Isabela and south of Marchena in the three zones suggested by Harris (1969) [5,8]. Since the within-island biological variability of this Central-Southeastern zone was larger than the between-island biological variability [5], local differences in environmental conditions, e.g., temperature and physical habitat characteristics, may indeed have been crucial to shape the observed fish assemblages. Nevertheless, studies that assess the effect of local environmental conditions have been few. Jennings et al. (1994) and Edgar et al. (2004) provided strong evidence for differences in fish assemblages, but they were unable to identify any relationships with environmental conditions due to lacking data [5,9]. Llerena-Martillo et al. (2018) combined visual census data with environmental measurements to study fish assemblages in mangrove ecosystems in Santa Cruz, but the number of sites and number of variables remained limited [10].

Besides natural stressors, anthropogenic pressures may also affect fish assemblages. During the last three decades, the Galapagos archipelago has been witnessing an increase in population, tourism, and waste production at an annual rate of 4.08, 6.71, and 4.02%, respectively, but good waste management procedures have not been evolving at the same pace [11–14]. Since pollution has been identified as a major threat to marine ecosystems [15], understanding the drivers behind the structure of fish assemblages in the Galapagos archipelago is crucial to evaluate and/or propose conservation guidelines [16].

The aim of this study was to identify the factors responsible for patterns in diversity and the structure of fish assemblages on the rocky shores of the Central-Southeastern zone of the Galapagos archipelago. To this end, the coastal areas of two cities, significantly different in size, on two different islands were assessed using underwater video transects. Difficulties in separating the effects of the potential drivers, i.e., dispersal limitation, water conditions and physical habitats, were partially circumvented by applying a multi-scale approach [17]. First, dispersal limitation can prevent fish to track and respond to environmental differences and, therefore, can affect ecologically processes at a larger, more

regional scale (>100 km) than environmental preferences [18–20]. Second, the wide range of water conditions in the archipelago, which are the result of distinct oceanic currents and urbanization, are more likely to affect communities at a more intermediate scale (0.1–100 km) [16,21,22]. Finally, the composition of physical habitats (e.g., sand and bare rock) seems to mainly vary at a relatively fine scale [23], causing it to potentially affect communities at a more local scale (1–100 m). A hierarchical multi-scale repeated-observations sampling design was used to collect biological data in the Central-Southeastern zone of the archipelago. These data were used to compare fish assemblages between islands, between locations and within locations.

Given the expected differences in variation and magnitude of anthropogenic pressures and environmental conditions between and within the islands of Santa Cruz and Floreana, we hypothesize that fish assemblage structure and diversity will differ accordingly and will be affected by different factors. We hypothesize that, in the highly populated bay of Santa Cruz, local anthropogenic pressures induce pronounced gradients in water conditions, which affect the structure and diversity of the fish assemblages. Fish diversity is expected to be negatively affected by human-induced changes of the water quality. In the sparsely populated bay of Floreana, we expect less pronounced gradients in water conditions and, therefore, a stronger effect of the composition of the physical habitat.

Although the main aim of this study was to determine the main factors that steer the fish assemblages of the Galapagos, the obtained results are also of direct use for the many other tropical oceanic islands that face increasing anthropogenic pressures.

## 2. Material and Methods

### 2.1. Data Collection and Sampling Design

Data collection included 540 observations distributed evenly among 10 locations on two islands. Two cities on two different islands within the Central-Southeastern zone of the Galapagos archipelago, which were expected to have significantly different levels of anthropogenic pollution, were selected: Puerto Ayora (Academy Bay) on Santa Cruz island and Puerto Velazco Ibarra on Floreana island. Puerto Velazco Ibarra is the smallest city of the archipelago with 111 inhabitants, while Puerto Ayora is the largest city with 11,822 inhabitants and the highest number of visiting tourists, according to a survey of 2015 [24]. Per city, five locations with rocky habitats were chosen along the coast to use fixed video transects to study fish assemblages. Video transects were chosen over traditional visual census techniques, as the former allows us to (i) store video data for later analysis, (ii) reduce the amount of time spent on field work and (iii) improve the standardization of data collection [25,26]. The fish data collection was based on a standard operation procedure developed by the Aquatic Ecology Research Group of Ghent University (see further). On each location, three transects with a length of 50 m each were laid out using ropes. For fish to be included, they had be recorded within 2.5 m of either side of the transect line (area = $50 \times 5 = 250$ m$^2$). Observers were trained to recognize whether fish had to be considered, depending on the estimated distance from the rope. All transects were monitored at a constant depth of 1.5 m, parallel to the coastline, and only locations with a limited exposure to waves were selected to guarantee the safety of the observers and to avoid predominant effects of wave exposure on the structure of fish assemblages. All transects were approximately 20 m apart. In practice, this meant that transects were placed next to each other along the coastline. To account for observer bias and sampling variability, each of these transects was recorded six consecutive times with single GoPro cameras (GoPro Hero 5 Black, 1080p, 60 fps, wide FOV) by three different observers equipped with a mask and snorkel. Hence, ten locations × three transects × three observers × six repeats yielded 540 observations (18 per transect) (Figure S1.1). In summary, two islands, with five locations each, were studied. In each location, three transects were each covered six times by each of the three observers.

The observers covered the transects in a browsing fashion, similar to the S-type transects introduced by Pelletier et al. (2011) [27]. Observers browse through the transect

at a fixed speed, but varying angle and can zoom in if needed to find individuals hiding in crevices. This S-transect was chosen over the more standarized I-transect with fixed angle and without zooming because the former had been found to enable detection of more individuals and species compared to the latter [27]. Transects were placed during low tide and monitored during flood tide of consecutive days from the 19 until the 31 of August 2017. The duration of each observation was approximately 4.6 min ($\pm$0.7 min). The time between each observation was at least one minute and subsequent observations were found independent in terms of diversity and assemblage structure (Section S2). Per day, one location was monitored. The analysis of the videos included species identification and an estimation of the number of individuals per species, determined as MaxCount, i.e., the total number of individuals per species per observation. The videos were analyzed once by one out of two video analysts; videos were assigned randomly to these analysts to reduce inter-observer bias [28].

Halfway the biological monitoring of a specific location, physical-chemical conditions were determined once using in situ measurements. In situ measurements were based on standard operation procedures developed in the Aquatic Ecology Research Group of Ghent University: Water temperature, acidity (pH), electrical conductivity (EC), dissolved oxygen (DO) and chlorophyll were measured with handheld multiprobes (WTW for temperature ($^\circ$C), pH (-) and DO concentration (mg L$^{-1}$); Aquaread (AP5000 version 4.07) for EC (mS cm$^{-1}$), and chlorophyll ($\mu$g L$^{-1}$)). Due to unstable measurements, chlorophyll concentrations were only used to compare average values of both islands and were not considered as a parameter in the constructed models. Nutrient concentrations of nitrite (mg NO$_2^-$-N L$^{-1}$), nitrate (mg NO$_3^-$-N L$^{-1}$), ammonium (mg NH$_4^+$-N L$^{-1}$), sulfate (mg SO$_4^{2-}$ L$^{-1}$), phosphate (mg PO$_4^{3-}$-P L$^{-1}$), total phosphorus (total P) (mg P L$^{-1}$), and total nitrogen (total N) (mg N L$^{-1}$) were determined spectrophotometrically (ThermoFisher Genesys 10S UV-VIS) using Merck kits. Measurements that were below the detection limit, were given the value of the detection limit. This was only the case for ammonium concentrations in Floreana and at location B10 in Santa Cruz, which were given a value of 0.01 mg NH$_4^+$-N L$^{-1}$. To assess how the anthropogenic influences differed between islands, in five locations in Floreana and three locations in Santa Cruz, the presence/absence of coliforms and *Escherichia coli* was determined as a measure of fecal contamination. To assess the underlying causes of patterns in water quality (i.e., variability in natural conditions or variability in anthropogenic pollution) satellite imagery and simulations from remote sensing based models were used. Results suggested that Floreana has lower levels of anthropogenic pollution but more clear signs of natural upwelling than Santa Cruz. More information is provided in Section S3.

For each transect, the percentage cover of different habitat classes (i.e., bare rock, rock with sediment deposition, vegetated rock and sand) was determined within 2.5 m of each side of the rope through visual inspection of the video data. Physical habitats are often complex and difficult to classify. Therefore, each transect was covered multiple times to assess how closely fish assemblages were associated with a specific (series of) micro-habitat(s). Since the scale and the intensity of the biological and environmental sampling differed, the biological data was aggregated to the level of transects (30 transects, i.e., 2 islands $\times$ 5 locations $\times$ 3 transects) to fit the scale of the abiotic data before applying constrained ordination. In addition, because of the spatially nested nature of the data, the geographical distances between sampling units (i.e., transects) were at significantly different scales (i.e., between islands, between locations and between transects). Therefore, similarities between nearby locations and transects may be the result of the spatial structuring of species distributions rather than local environmental conditions [29–31]. This spatial structuring, which manifests itself in the data as spatial auto-correlation, was accounted for by explicitly integrating the geographical distance between the sampling units in the different models [31].

To aggregate the biological data within transects the median of the 18 observations per transect was chosen over the mean or maximum. Since fish tend to be very mobile,

occasional observations of different species moving between locations, may yield questionable results. Using the median rather than the mean accounts for this bias, though it comes at a cost, since individuals of cryptic, rare, and easily scared species will also be less likely to be observed during multiple observations.

### 2.2. Data Analysis

All analyses were performed using the R software [32] and the Primer v6 multivariate statistics package [33] with PERMANOVA add-on [34]. In Figure S1.2, the data analysis roadmap is presented.

### 2.2.1. Spatial Variability of Water Conditions and Habitats

In this study, geographical distance, i.e., normalized longitude (X) and latitude (Y), was used to account for spatial auto-correlation and dispersal. The distances between the locations within one island were negligible compared to the distance between the two islands. Despite the aim of selecting locations at similar distances of each other, due to logistic constraints this was not always possible. In Floreana, the locations were more or less at equal distances from each other along the coast, but this was not the case for Santa Cruz, where locations were positioned within a bay, spatially clustering locations B2 and B3, as well as B9 and B10, while B6 was more isolated (Figure 1). The average distance between locations for Santa Cruz and Floreana was $734 \pm 544$ m (SD) and $525 \pm 397$ m (SD), respectively.

The DISTance-based Linear Model (DISTLM) (see further) does not make any assumptions about the distributions of the covariables, but these distributions should nevertheless be appropriate for linear modelling [34,35]. Therefore, to assess whether distributions of covariables were not skewed or contained outliers, draftsman plots were constructed and analyzed. To deal with the detected skewness and outliers of some of the covariables, these were transformed before normalization (subtraction of mean and division by standard deviation). Nitrite, nitrate, ammonium, total N, total P, phosphate, and DO concentration were log(V) transformed and sulfate concentration was -log(2-V) transformed. Similarly, the cover of different habitat classes was often left or right skewed. Therefore, the percentage cover of the physical habitats was arcsine square root transformed prior to conducting the analyses [36]. The resulting histograms and residual plots indicated a better fit compared to the case of no transformation (tested for all covariables) or logit transformation (tested for the cover of the habitat classes).

Principal Component Analysis (PCA) was used to visually assess any potential grouping of the water variables and the physical habitats. Tests of homogeneity of dispersions (PERMDISP) based on Euclidean resemblance matrices of the physical habitats and water conditions allowed to determine and compare the variability of both series of environmental variables in both islands. Additional PERmutational Multivariate dissimilarity-based ANOVA (PERMANOVA) analyses allowed to assess whether physical habitats and water conditions were significantly different between islands and locations (nested in islands). The PERMANOVA analyses were done using $10^5$ unrestricted permutations of the raw data. To compare the physical habitats of the islands, locations were considered as a random factor. To compare the physical habitats of the locations, one analysis was performed for both islands together. If both PERMANOVA and PERMDISP tests were found to be significant, additional Canonical Analysis of Principal coordinates (CAP) tests were used to assess the distinctiveness of the considered groups.

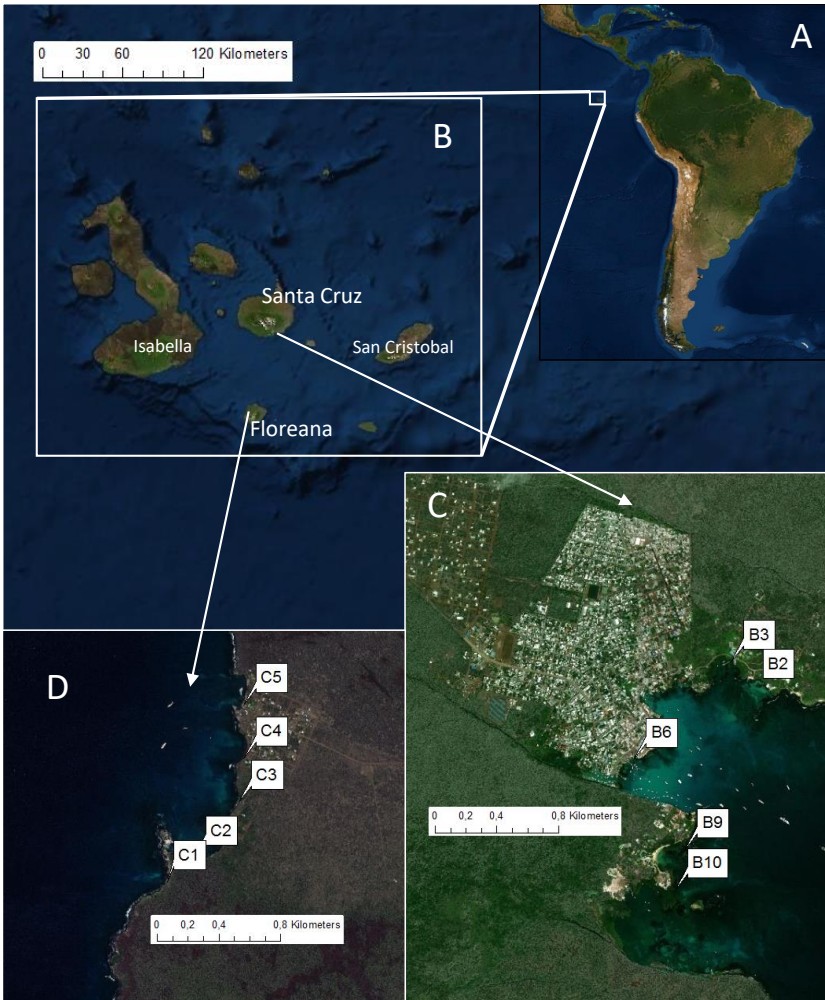

**Figure 1.** Map of the study area. (**A**) South-American continent with depiction of the Galapagos archipelago. (**B**) Galapagos archipelago with depiction of the two studied islands. (**C**) The city Puerto Ayora on Santa Cruz island with indication of the study locations. (**D**) The city Puerto Velazco Ibarra on Floreana island with indication of the study locations. Landsat 8 imagery was used to construct the maps.

### 2.2.2. Fish $\alpha$ and $\beta$ Diversity

In literature, there have been many different definitions for both $\alpha$ diversity and $\beta$ diversity, underlining the importance of a clear description of the definition used in each study. In this study, we mainly adopted the definitions provided by Gray (2000) [37], which are closely related to the original definitions provided by Whittaker (1975) [38]. The number of species observed per sampling unit is referred to as point species richness. Since it is unlikely that only one sampling unit would be representative for the entire study area, a sample typically comprises multiple sampling units within the area. The number of species found within a sample is referred to as sample species richness, which is more closely related to Whittaker's $\alpha$ diversity. In this study, the number of species observed during a single observation is defined as the point species richness, while the total number of species within an island, location or transect is referred to as the sample species richness. The sample species richness was assessed visually using Species Accumulation Curves (SACs) ($10^5$ permutations) for the different islands (n = 270 per island) and locations (n = 54 per location) (Section S10.1). SACs depict the number of observed species with increasing sampling effort and are typically used to estimate the number of species in a particular area [39]. To statistically compare the sample species richness of islands and locations (Wilcoxon rank sum tests) and to assess correlations with environmental variables,

respectively, the total number of observed species per location and transect (maximum sample species richness) were retained for analysis. To compare the point species richness of both islands, a linear mixed model with island as fixed effect, location and transects as nested random effects and observer as crossed random effect were constructed. To compare the point species richness of the locations, one analysis was performed for both islands together using a linear mixed model with location as fixed effect and transect and observer as nested and crossed random effects, respectively.

Although $\beta$ diversity can be defined in numerous ways, in this study it is defined as the variability in species composition among samples for a given area at a given spatial scale [38,40,41]. Differences in $\beta$ diversity at the spatial scale of (1) islands and (2) locations were assessed using a PERMDISP test on the Sorensen resemblance matrix of the samples, in which a sample corresponds with the aggregated data of a single transect. The distance to centroid in a two-dimensional Principal Coordinate (PCO) space is considered a measure of $\beta$ diversity [40]. To compare (1) islands and (2) locations, the centroids of the samples of (1) each island and of (2) each location were used, respectively. To assess if there were any relationships between environmental conditions and diversity, Pearson correlation coefficients (*r*) of the diversity measures versus the transformed environmental variables were determined.

### 2.2.3. Fish Assemblage Structure

Differences in the structure of fish assemblages between islands are not necessarily the result of their position along an environmental gradient, but rather the result of fragmentation due to migration barriers, local extinction or colonization [42,43]. Therefore, parametric methods, such as Multivariate dissimilarity-based ANOVA (MANOVA), PCA, CCA, and RDA, which assume an underlying environmental model of species distributions [44], are not appropriate. In addition, ecological data, especially those originating from visual census, are often overdispersed and zero-inflated, limiting the use of traditional parametric methods. On the other hand, purely non-parametric approaches, such as ANOSIM, are unable to partition variability, assess interactions or model more complex patterns [34]. Semi-parametric multivariate techniques, such as PCO, PERMANOVA, CAP, and distance-based Redundancy Analysis (dbRDA), combine the advantages of both approaches and rely on the actual values of the resemblance matrix instead of relative ranks, but still use permutations rather than distributional assumptions [34]. Nevertheless, by using the actual dissimilarity values instead of ranks, the choice of transformation, aggregation and measure of resemblance become more important [34].

The analyses were performed on the Bray-Curtis resemblance matrix of the original fourth-root transformed biological data set. Prior to applying PERMANOVA, a PERMDISP analysis was performed to test for homogeneity among the dispersions of the different islands and locations [45] (Section S10.2). Although PERMANOVA is quite robust to heterogeneity of variances for balanced designs when compared to other methods, such as ANOSIM or the Mantel test [46], significant differences in multivariate variability may complicate interpretations of PERMANOVA analyses. More specifically, if both PERMDISP and PERMANOVA yield significant results, there is not necessarily a significant discrepancy in the structure of the fish assemblages other than the difference in variability. Therefore, additional CAP discriminant analyses were applied to assess the distinctiveness in the structure of fish assemblages between the islands, locations and transects. The leave-one-out misclassification error was used as a measure of the distinctiveness of each of the groups.

To compare islands using PERMANOVA, the considered factors were island (fixed), location (random and nested within island), transect (random and nested within location) and observer (random and crossed). To compare locations using PERMANOVA, irrespective of the islands, the factor island was dropped and the factor location was treated as the fixed factor. The assessment of the main effects and pair-wise comparisons were obtained using $10^4$ permutations of residuals under a reduced model. The square root components of

variation ($\sigma'$) were estimated by equating the mean squares of the PERMANOVA models to their expectations [34]. To determine the proportion of the variability explained by the different grouping factors, the percentage of variation of each component to the total variation was used. PCO analyses were performed to visualize the multivariate patterns. The fish species that were characteristic for the differences among the islands and locations were found by superimposing vectors corresponding to Pearson correlations of individual species with the resulting PCO axes.

CAP analyses were also performed to describe the correlation of the aggregated multivariate biological data to gradients of water variables, physical habitats and geographical distance. The first PCA axis, describing each series of variables, being habitat, water conditions and distance, was retained and plotted against the first CAP axis of the aggregated biological data set and the corresponding canonical correlation of both axes was determined.

To assess the relative importance and overlap of geographical distance, water conditions and physical habitats in shaping the observed fish assemblages, they were considered as separate subsets of variables in DISTance-based Linear Models (DISTLMs). While CAP finds linear combinations of the biotic and abiotic variables that are maximally correlated with one another, DISTLM finds linear combinations of the abiotic variables that are best to predict patterns in the biotic data set [34]. As such, DISTLM models take into account the potential overlap of different predictors. Data was aggregated to the level of transects (median) and spatial auto-correlation was accounted for by explicitly integrating geographical distance. Because the number of spatial variables was limited compared to the number of variables of the other two series, the polynomials up to 3rd order of the coordinates, i.e., X (longitude) and Y (latitude), were included, as well. Since the number of variables exceeded the amount of observations, no interactions were considered and subsets of variables were established for each series using forward selection based on the multivariate analogue to the small-sample-size corrected version of the Akaike Information Criterion (AICc) and sequential conditional distance-based Redundancy Analysis (dbRDA). To compare the series, ideally the number of variables per subset should be the same. However, the optimal number of variables based on the AICc can differ between the series. Therefore, the effect of including fewer or more variables was assessed and the variability of the results was evaluated (Section S9). These subsets were then compared using the AICc values, forward selection and sequential conditional dbRDA. Models were also constructed for all variables independent of the series they were assigned to, using forward selection. Finally, this approach was extended by assessing all possible combinations of variables to construct q-variable models with q ranging from one to six.

It is important to note that for the analyses that only considered biological data (i.e., PERMDISP, PERMANOVA, and CAP) the full biological data set was used (n = 540), while, for the analyses that considered both environmental and biological data (i.e., DISTLM and CAP-PCA), the aggregated biological data set was used (n = 30).

## 3. Results

### 3.1. Spatial Variability of Water Conditions and Habitats

There were significant differences between both islands ($p < 0.05$; Wilcoxon rank sum test) in some of the measured water variables (Table S4.1). In Floreana, nitrite and sulfate concentrations were significantly higher, while, in Santa Cruz, ammonium concentrations, temperature, pH, and DO concentrations were significantly higher. Although not significant, nitrate, phosphate, total P and chlorophyll concentration were on average higher in Santa Cruz than in Floreana, while total N and EC were on average higher in Floreana. Although rocky habitats were targeted for the transects, sandy patches were often unavoidable and sand cover was significantly higher in Santa Cruz (the abiotic data is depicted visually in PCA plots (Figures S4.1–S4.3). PERMANOVA tests using all available water variables also highlighted significant differences in water quality between both islands ($p < 0.05$), and the lack of significant differences in distances to the centroids

(PERMDISP tests) suggests that the significant PERMANOVA was due to true differences in multivariate water quality data rather than to differences in variability. However, when PERMANOVA was performed only on the selection of water parameters identified in Section 3.3 as potential drivers of the structure of fish assemblages (temperature, ammonium, phosphate, nitrite and sulfate concentration), both PERMANOVA and PERMDISP yielded significant results, showing a higher multivariate dispersion among locations in Santa Cruz than in Floreana and potentially questioning a significant island effect on the abiotic environment. However, the CAP leave-one-out cross-validation misclassification error was exactly 0%, suggesting that both islands could still be perfectly distinguished from each other based on their water parameters.

Similarly, PERMDISP tests indicated a significantly larger variability of the physical habitats among locations in Santa Cruz compared to Floreana. Here, PERMANOVA tests were only marginally significant ($p < 0.1$), suggesting that there were no clear differences in habitat classes between both islands, in addition to the difference in variability. This was supported by the high CAP leave-one-out cross-validation misclassification error (20%; note that a misclassification error of 50% is expected by chance alone). Between locations, no significant differences were found for both PERMDISP and PERMANOVA.

Coliforms and *E. coli* were found more often in Santa Cruz than in Floreana (Table S4.2). In three out of three locations in Santa Cruz, both coliforms and *E. coli* were found, while, in Floreana, the respective frequencies were three out of five and one out of five locations.

### 3.2. Fish α and β Diversity

The linear mixed models did not indicate any significant difference in point species richness between both islands ($p > 0.05$). Similarly, Wilcoxon rank sum tests did not indicate any significant difference in sample species richness. The total amount of species recorded in Santa Cruz and Floreana was 39 and 33, respectively (Figure S5.1). There was a significant effect of the factor Location in the linear mixed models for the locations, irrespective of the islands. Pairwise comparisons with Tukey correction of these models revealed significantly higher point species richness in location B9 compared to locations C1, B3 and B6; and in location C4 compared to location B3.

β diversity among locations was significantly larger in Santa Cruz than in Floreana ($p < 0.05$), indicating stronger variability in species composition in Santa Cruz (Figure S5.2). The distance-to-centroid for Santa Cruz was on average 31.8% (SE of 2.4%), while, for Floreana, it was only 17.5% (SE of 2.4%). Pairwise comparisons between locations, without correction for multiple comparisons, did only result in some marginally significant differences ($p < 0.1$) in β diversity. However, it should be noted that the number of transects per location was limited to three, impeding strong statements on potential differences between locations due to limited statistical power. The range of distance-to-centroid values was quite large, from 3.82% to 27.63%, but more data is required to assess the β diversity at this fine spatial scale (Table S5.4 and Figure S5.2).

There was only a limited amount of significant correlations between diversity measures and environmental variables. Nitrite concentration was negatively correlated with the sample species richness of locations in Floreana ($r = -0.96$). In Santa Cruz, Total P concentration ($r = -0.94$) and habitat class sediment deposition on rocks ($r = -0.91$) were negatively correlated with point species richness, while EC was positively correlated ($r = 0.94$) with point species richness (Table S5.2). Although not all types of nutrient concentrations showed significant correlations with point species richness and sample species richness, correlations between nutrient concentrations and point species richness and sample species richness were always negative in Santa Cruz. In Floreana, on the other hand, there were both positive and negative non-significant correlations between nutrient concentrations and point species richness and sample species richness. In Floreana, β diversity was positively correlated with nitrate ($r = 0.91$) (Table S5.3). In Santa Cruz, β diversity was negatively correlated with DO ($r = -0.92$) and sand cover ($r = -0.88$).

A positive, but not significant ($p = 0.13$), Pearson correlation ($r = 0.52$) was found between $\beta$ diversity and physical habitat variability (Table S5.4).

### 3.3. Fish Assemblage Structure

PERMDISP tests on the full biological data set indicated significant heterogeneity of multivariate dispersion at the level of islands, locations, and transects ($p < 0.05$), but not at the level of the observers. Multivariate variability was significantly larger for Santa Cruz than for Floreana, as the average distance-to-centroid of Santa Cruz was 35.44% (SE = 0.48%), while, for Floreana, it was only 27.30% (SE = 0.49%). PERMANOVA models revealed significant effects at the level of islands and locations ($p < 0.05$). Within the island of Santa Cruz, more significant differences ($p < 0.05$) were found between locations (B10-B3; B10-B6; B10-B9; B3-B9; B6-B9) than between the locations of Floreana (C1–C4). The *p*-values were not corrected since the permutation *p*-values provide an asymptotically exact test of each individual null hypothesis of interest [34] (Section S10.3). Because PERMDISP and PERMANOVA analyses both yielded significant results, there was not necessarily a significant difference in the structure of the fish assemblages in addition to the difference in variability. However, CAP discriminant misclassification errors were generally low, indicating that the locations and transects of both islands could be relatively well distinguished from each other. Hence, the structure of the fish assemblages differed between islands, locations and transects. Differences among locations and transects were clearer (i.e., had lower misclassification errors) at Santa Cruz than at Floreana (Tables S8.1–S8.4). These results are clearly visualized in the PCO plot, where the fish assemblages of both islands were clearly separated (Figure 2), and where there is much more overlap among locations at Floreana than at Santa Cruz (Figures S7.1 and S7.2). In addition to the factors Island (Table S6.1: 57.09%) and Location (12.47%), the factor Transect also explained a substantial part of the variation (12.74%), whereas the factor Observer alone explained a negligible fraction of the data variation (0.27%). The interaction of Observer with the factors Island (0.06%), Location (2.97%), and Transect (4.39%) still never explained more than 7.5% of the observed variation. In Santa Cruz, variability explained by the locations (38.05%) was larger than the variability explained by the transects (28.72%), while the opposite was true for Floreana (i.e., locations and transects explained 16.62% and 17.60% of the variation, respectively).

In both Santa Cruz and Floreana, Endemic, Peruvian, Indo-Pacific, Panamic, and widespread species were observed (Table S5.1). Using the correlations of the biological data with the first PCO axis, the differences between islands could be partly attributed to some typical species. For Santa Cruz, these included the Bullseye puffer fish (*Sphoeroides annulatus*; $r = -0.62$), Yellowtail damselfish (*Microspathodon bairdii*; $r = -0.83$) and Pacific spotfin mojarra (*Eucinostomus dowii*; $r = -0.64$), while, for Floreana, they comprised the Bravo clinid (*Gobioclinus dendriticus*; $r = 0.56$), Galapagos ringtail damselfish (*Stegastes beebei*; $r = 0.83$) and Chameleon wrasse (*Halichoeres dispilus*; $r = 0.87$). In Santa Cruz, the abundances of the Yellowtail damselfish and Pacific spotfin mojarra were positively correlated with DO and pH, while the abundances of the Bullseye puffer fish were positively correlated with phosphate and negatively correlated with EC. In Floreana, the abundance of the Galapagos ringtail damselfish was positively correlated with nitrite, while that of the Bravo clinid was negatively correlated with the percentage of bare rock and that of the Chameleon wrasse was positively correlated with phosphate.

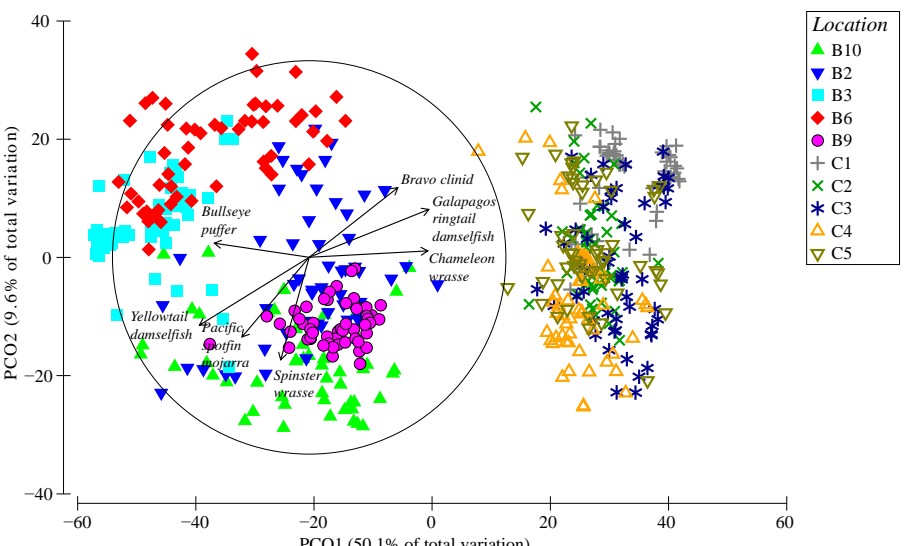

**Figure 2.** Principal Coordinate (PCO) based on Bray-Curtis resemblance matrix with distinction of the different locations in Santa Cruz (locations B) and Floreana (locations C). The fish species with a Pearson correlation of more than 0.4 with the ordination axes are represented as vectors.

The squared canonical correlations of PC1 with CAP1 indicate how well differences in the structure of fish assemblages can be explained by the water conditions, physical habitats and geographical distance. Compiling data of both islands, these correlations (n = 30) were quite high: 0.85, 0.66, and 0.90, respectively (Table 1). At the level of individual islands, the squared correlations were 0.75, 0.76, and 0.36 for Santa Cruz (n = 15), while, for Floreana (n = 15), they were much lower: 0.03, 0.28, and 0.16, respectively.

When focusing on water conditions, the most parsimonious model that best explained fish community variation for both islands contained five variables: Temperature, ammonium, phosphate, nitrite and sulfate (q = 5). When focusing on the habitats, three variables were retained in the most parsimonious model: Cover of vegetated rock, bare rock and rock with sediment deposition (q = 3). When focusing on the geographical distance, Y and $X^3$ were retained in the most parsimonious model (q = 2). For Santa Cruz, total P and temperature (q = 2), cover of rock with sediment deposition (q = 1), and $Y^3$ X and Y (q = 3) were retained. For Floreana, nitrite (q=1), cover sand and bare rock (q = 2) and $X^3$ (q = 1) were retained (Table S9.1).

For the full data set, water conditions explained the largest share of the variation, followed by the geographical distance and physical habitats (marginal tests in Table **??**). The sequential conditional tests indicated that the most parsimonious model contained only geographical distance. If geographical distance was not included, the most parsimonious model only comprised water conditions. Changing the number of variables per subset of each series (i.e., geographical distance, water conditions and physical habitats) did not change the order of importance of the series but did affect the number of series to be included in the parsimonious model (Table S9.3).

Within each island, geographical distance was not included in the most parsimonious models (Table **??**). Indeed, the sequential conditional tests for Floreana only retained physical habitats to explain the structure of the fish assemblages, while, for Santa Cruz, only water conditions were retained in the most parsimonious model. However, for Santa Cruz, the AICc of the model using solely geographical distance was only slightly higher ($\Delta$AICc < 2). In addition, care should be taken when comparing these results, because the number of variables included in the subsets had an effect on the order and nature of the series retained in the parsimonious model of Santa Cruz (Table S9.2). If one instead of two variables per series were included, both physical habitat and geographical distance

were added to the parsimonious model, but water conditions were left out (Table S9.2). For Floreana, there was no effect of the number of variables on the order or nature of included series. To visualize the outcomes of these models ordinations of their fitted values are represented in Figures S9.1, S9.3, S9.4, S9.6, and S9.7. As the main patterns of the unconstrained PCO ordination plots (Figures S9.2, S9.5, and S9.8) and constrained dbRDA plots are relatively similar (comparing original values and fitted values, respectively [34]), the models appear adequate to find and explain the most important patterns in the data.

**Table 1.** Squared canonical correlations of the first Principal Component Analysis (PCA) axis (PC1) with the first Canonical Analysis of Principal (CAP) axis (CAP1) for the different series of variables: Water parameters (Water), physical habitats (Habitat), and geographical distances (XY) for both islands, together (n = 30) and separately (n = 15).

| Variable Series | Both Islands | Santa Cruz | Floreana |
|---|---|---|---|
| Water | 0.85 | 0.75 | 0.03 |
| Habitat | 0.66 | 0.76 | 0.28 |
| XY | 0.90 | 0.36 | 0.16 |

**Table 2.** Results of DISTance-based Linear Model (DISTLM) analyses using series of predictors (water parameters (Water), physical habitats (Habitat) and geographical distances (XY)). The optimal number of predictors per series was determined using the Akaike Information Criterion (AICc). For each marginal test, the AICc weights ($w_{AICc}$) were also determined. Models were constructed for both islands, together (n = 30) and separately (n = 15).

| | Variable Series | Marginal Tests | | | | | Sequential Tests | | |
|---|---|---|---|---|---|---|---|---|---|
| | | Pseudo-F | *p*-Value | $R^2$ | AICc | $w_{AICc}$ | Pseudo-F | *p*-Value | Cum. $R^2$ |
| Both islands | XY (q = 2) | 21.43 | 0.001 | 0.61 | 202.11 | 0.73 | 21.43 | 0.001 | 0.61 |
| | Water (q = 5) | 10.71 | 0.001 | 0.69 | 204.18 | 0.26 | / | / | / |
| | Habitat (q = 3) | 9.03 | 0.001 | 0.51 | 211.89 | 0.01 | / | / | / |
| Santa Cruz | Water (q = 2) | 5.52 | 0.001 | 0.48 | 103.17 | 0.48 | 5.52 | 0.001 | 0.48 |
| | XY (q = 3) | 4.95 | 0.001 | 0.58 | 103.96 | 0.32 | / | / | / |
| | Habitat (q = 1) | 4.91 | 0.001 | 0.27 | 104.97 | 0.20 | / | / | / |
| Floreana | Habitat (q = 2) | 4.00 | 0.001 | 0.40 | 94.46 | 0.66 | 4.00 | 0.001 | 0.40 |
| | XY (q = 1) | 1.74 | 0.144 | 0.12 | 97.06 | 0.18 | / | / | / |
| | Water (q = 1) | 1.56 | 0.187 | 0.11 | 97.24 | 0.16 | / | / | / |

When no prior clustering of predictors in series (water conditions, physical habitats, or geographical coordinates) was done, a parsimonious model for the full data set contained the predictors Y, cover of rock with sediment deposition, temperature, ammonium concentration, and cover of vegetated rock (Table **??**). For Santa Cruz, a two-variable model containing cover of rock with sediment deposition and temperature was considered best, while, for Floreana, a two variable model containing cover of sand and bare rock was optimal.

**Table 3.** Results of DISTLM analyses using all predictors, without grouping them in different series of variables. Predictors were selected using AICc. For both islands, together (n = 30) and separately (n = 15).

| | Variable | Marginal Tests | | | Sequential Tests | | | |
|---|---|---|---|---|---|---|---|---|
| | | Pseudo-F | *p*-Value | $R^2$ | AICc | Pseudo-F | *p*-Value | Cum. $R^2$ |
| Both islands | Y (latitude) | 32.81 | 0.001 | 0.54 | 204.89 | 32.81 | 0.001 | 0.54 |
| | Rock with sediment deposition | 9.08 | 0.001 | 0.24 | 201.96 | 5.33 | 0.001 | 0.62 |
| | Temperature | 19.37 | 0.001 | 0.41 | 200.09 | 4.25 | 0.001 | 0.67 |
| | Ammonium | 17.45 | 0.001 | 0.38 | 199.26 | 3.31 | 0.004 | 0.71 |
| | Vegetated rock | 9.95 | 0.001 | 0.26 | 199.18 | 2.73 | 0.012 | 0.74 |
| Santa Cruz | Rock with sediment deposition | 4.91 | 0.001 | 0.27 | 104.97 | 4.91 | 0.002 | 0.27 |
| | Temperature | 4.00 | 0.003 | 0.24 | 102.66 | 5.30 | 0.002 | 0.50 |
| Floreana | Sand | 3.94 | 0.003 | 0.23 | 94.97 | 3.94 | 0.004 | 0.23 |
| | Bare rock | 3.31 | 0.011 | 0.20 | 94.47 | 3.34 | 0.023 | 0.40 |

## 4. Discussion

Although the $\alpha$ diversity, $\beta$ diversity, and structure of the fish assemblages were clearly different between both islands, elucidating the drivers of these differences is not evident because of the complex interplay of often intercorrelated environmental variables and spatially confounded factors (e.g., occurrences of natural upwelling and anthropogenic pollution) [47,48]. It is, however, advisable to consider such a large set of potential drivers as other metrics to characterize fish assemblages of similar islands in the Tropical Eastern Pacific have been found to be affected by multiple different environmental, biogeographical and anthropogenic factors [6,49,50]. Nitrite, ammonium, sulfate, DO concentration, temperature, and pH were significantly different between both islands and sand cover was significantly higher within the transects of Santa Cruz. In addition, variability of the water conditions and physical habitats were both larger in Santa Cruz than in Floreana. Furthermore, Santa Cruz was characterized by strong anthropogenic pressures and little natural upwelling, while the contrary was true for Floreana. Because the series of variables were characterized by many correlations among their variables, there was no sound statistical ground to separate the effects of geographical distance, water quality and physical habitats on diversity and the structure of fish assemblages based on this observation alone. However, by collecting data over multiple scales, i.e., islands, locations, and transects, a more detailed assessment could be made [51].

### 4.1. Diversity

Although the point species richness and sample species richness did not seem to differ between the islands, the total number of species and $\beta$ diversity in Santa Cruz were higher than in Floreana. This suggests that on a habitat scale, i.e., within transects and within locations, the number of species may have been the same, but that the variability in species composition was larger in Santa Cruz than in Floreana. This is expected as the variability in observed water conditions and physical habitats of the different locations was also higher in Santa Cruz than in Floreana, and more different niches are more likely to harbor more different species [52,53]. In addition, the $\beta$ diversity did not differ significantly between the locations, which corroborates the absence of significant differences in the variability of physical habitats between the locations. However, it should be noted that the number of transects per location, and, therefore, the statistical power, was limited. The fact that the $\beta$ diversity of locations showed a weak positive correlation with physical habitat variability and that $\beta$ diversity of the islands seemed positively affected by the variability of water

conditions and physical habitats, suggests that environmental conditions may be important for diversity on multiple spatial scales. Nevertheless, more fine-scale studies are required.

In Santa Cruz, lower levels of $\alpha$ diversity (both point and sample species richness) were associated with higher nutrient concentrations, which is in line with the results of meta-analyses on the effects of anthropogenic pollution on marine communities [54]. Nitrate and phosphate concentrations measured during this study in Santa Cruz were found to be four and two times higher than those recorded in June 2007 [55] and eight and seven times higher than those recorded in 1968 [21], underlining the increasing risk fish communities are facing. In addition, Mateus et al. (2019) found some strong relationships between population increase and nutrient concentrations in coastal sites of Santa Cruz during a 9-year study period [16]. However, whether the underlying cause behind elevated nutrient concentrations is natural or man-made remains often hard to determine. This is especially the case for the hydrodynamically complex area of the Galapagos archipelago, being subject to local upwelling of cold, saline, nutrient rich waters originating from the eastward Cromwell current [16,21,56]. A combination of satellite imagery, remote sensing based models and in situ nutrient measurements indicated that Floreana was characterized by strong natural upwelling, while high nutrient concentrations in Santa Cruz were more likely the result of anthropogenic pollution (Section S3.3). This result corroborates results of earlier studies [16,57]. Furthermore, the fact that fecal coliforms and *E. coli* were found more often in the coastal waters of Puerto Ayora, indicates a stronger anthropogenic pollution compared to Floreana [11,58].

### 4.2. Structure of Fish Assemblages

#### 4.2.1. Characteristic Species

DO and pH were significantly higher in Santa Cruz and seemed to have a positive effect on the abundance of the two most typical species from Santa Cruz, the Yellowtail Damselfish and the Pacific Spottedfin Mojarra. Hence, the absence of these species in Floreana may be related to the lower DO and pH. However, the abundances of the two most typical species of Floreana, the Galapagos Ringtail Damselfish and Chameleon Wrasse, were correlated only with nitrate and phosphate gradients in Floreana, respectively. These variables were not significantly different between the islands, hence providing no indication that environmental conditions are responsible for the relative absence of these species in Santa Cruz. However, it should be noted that the number of monitored locations per island was limited and that the measured environmental gradients in Floreana were less pronounced than in Santa Cruz.

#### 4.2.2. Fish Assemblages

The results suggest that the prevailing environmental conditions, rather than the geographical distance between the islands, are the underlying cause for the observed differences in the structure of the fish assemblages. First, in Santa Cruz and Floreana, a mixture of Endemic, Peruvian, Indo-Pacific, Panamic, and widespread species were found, confirming the conclusion of Edgar et al. (2004) regarding the strong connectivity of the archipelago with the surrounding islands and coasts [5]. Second, since, within the islands of Santa Cruz and Floreana, temperature and the cover of sand turned out to be most important for the structure of fish assemblages, and given the significant differences of these variables between both islands, it is likely that environmental conditions, rather than geographical distances, are the drivers behind the observed differences in the structure of the fish assemblages of both islands. In addition, the significantly stronger between-locations variability of the structure of the fish assemblages in Santa Cruz compared to Floreana, is likely the result of the pronounced environmental differences between the locations of Santa Cruz compared to those of Floreana.

Although multiple studies on fish assemblage structures of tropical oceanic islands have highlighted the importance of the degree of isolation, there have been conflicting results and even indications that other, often confounding, factors are at least equally

important [6,49,59]. Due to its unique geographical position at the intersection of multiple oceanic currents and the resulting wide range of environmental conditions, the Galapagos archipelago provides a unique case to assess the importance of these other factors. Indeed, despite its comparable degree of geographical isolation, the Galapagos has a relatively high species richness and functional dispersion compared to other tropical oceanic islands [6,60]. The importance of the wide range of environmental conditions, in a relatively small area, for the fish assemblage structure had already been suggested in other studies [5,61], but this is the first study to provide quantitative results in favor of this hypothesis.

Fish assemblages in Santa Cruz, an island characterized by anthropogenic pollution [16,57], are affected more by water conditions than physical habitats. The contrary is true for Floreana, where anthropogenic pollution is limited, invoking smaller gradients in water conditions (especially for those variables that turned out to be potential predictors for the structure of fish assemblages). To predict the structure of fish assemblages, nutrient concentrations, i.e., Total P, and temperature gradients were most important in Santa Cruz, while, in Floreana, the percentage sand cover was most important. Although sand cover within the observed transects was relatively low in Floreana, significantly different fish assemblage structures were observed along transects with high versus low sand cover.

Although the most parsimonious model for both islands only contained the geographical distance, latitude and longitude provided clearly a better representation of the geographical distance than ammonium, temperature and all other measured variables could ever provide for the environmental state of the water. Despite being inherent to any spatial environmental study, this limitation can be partially accounted for by including multiple spatial scales in the sampling design, as was illustrated here. As such, the main drivers behind the observed regional and local differences in fish assemblages could be identified with some level of statistical confidence, adding to the work of Harris (1969), Jennings et al. (1994), and Edgar et al. (2004) [5,8,9]. As suggested by these authors, temperature was identified as a major driver of fish assemblage structure. However, nutrient concentrations and the characteristics of the prevailing physical habitats play an additional important role.

## 5. Conclusions and Implications for Management

Differences in fish assemblage structure between Santa Cruz and Floreana are more likely the result of different water conditions rather than geographical distance or different physical habitats. However, while the fish assemblages in Santa Cruz were mainly affected by water quality, fish assemblages in Floreana were mainly affected by the type of physical habitats. This difference was assigned to the different levels of anthropogenic pressure on the coastal waters of both islands, with Floreana having less anthropogenic pressures and a more pristine nature than Santa Cruz. In Santa Cruz, higher levels of pollution seem to correspond with a lower $\alpha$ diversity and significantly different structure of fish assemblages, highlighting the importance of environmental and waste management in the populated bays of the Galapagos archipelago and confirming the negative effects of anthropogenic pressures on fish assemblages of tropical oceanic islands [1,6].

The structure of fish communities can be used as an indicator of human pollution. However, care should be taken as anthropogenic pressures and natural stressors can easily be confounded. Therefore, when assessing coastal water quality and anthropogenic pressures, naturally occurring local and seasonal variations should be taken into account. Although the sampling effort was distributed in such a way to provide reliable estimates of the underlying environmental drivers of fish assemblage structure and diversity, separating anthropogenic from natural stressors remained difficult. In this study, only anthropogenically influenced areas, with distinct natural stressors, e.g., temperature, were studied. Future studies should include more islands and stronger gradients of anthropogenic pressures to confirm the found results. In addition, estimates of fishing pressures and boat traffic should be used in future studies to provide a more complete assessment of the effect of human presence on the system. Although there have been several studies on the

water quality of coastal cities of the Galapagos and how it is related to population size and the number of incoming tourists, the pressures themselves (i.e., agricultural runoffs and untreated wastewater discharge) remain poorly understood. Mateus et al. (2019), for example, suggested that elevated phosphorus concentrations in the coastal waters of Santa Cruz could have been the result of inland pollution and water discharge, but studies to confirm this or to provide a more detailed description of the pollution sources and pathways remain absent [16].

This study can serve as baseline for future studies aiming to improve ecological understanding and/or develop environmental management guidelines of the Galapagos archipelago and other tropical oceanic islands that face similar threats of increasing anthropogenic pressures. Future studies should focus on the development of water quality criteria adapted to local conditions, identification of pressures on the environment, and management prioritization of pressures and associated environmental changes with the strongest effect on local ecosystems.

**Supplementary Materials:** The following are available online at https://www.mdpi.com/2077-1312/9/4/375/s1, S1: Sampling design and data processing and modelling roadmap, S2: Independence of observations, S3: Remote sensing, S4: Environmental analyses, S5: Fish $\alpha$ and $\beta$ diversity, S6: PERMANOVA, S7: PCO analysis, S8: CAP analysis, S9: DISTLM analysis, S10: Comments statistical analyses.

**Author Contributions:** S.B., R.B., P.G.: Conceptualization, S.B.: Data curation, Visualization, Writing—original draft, Formal analysis, Software, R.B., L.D.-G., P.G.: Funding acquisition, Supervision, Project administration, W.V.E., R.V., C.V.d.h., N.D.S., M.A.E.F., R.B., L.D.-G.: Investigation (water conditions), S.B., L.H., H.R., A.S., N.D.T., R.S., J.P.-C., P.G.: Investigation (fish assemblages), S.B., R.B., P.G.: Methodology, N.D.S., R.B., L.D.-G., P.G.: Resources, All authors: Writing—review and editing. All authors have read and agreed to the published version of the manuscript.

**Funding:** The research was partly funded by VLIR-UOS Biodiversity Network Ecuador, Special Research Fund of UGent and VLIR-UOS Global Minds. Travelling scholarships for staff and students were provided by FWO, CWO (UGent), VLIR-UOS (Global Minds), and VLIR-UOS Biodiversity Network Ecuador.

**Institutional Review Board Statement:** All methods were carried out in accordance with the relevant guidelines and regulations of the Galapagos National Park Directorate under research permit PC-02-19. All experimental protocols were reviewed and approved by the Galapagos National Park Directorate Applied Research Department, which assesses animal care in research activities.

**Informed Consent Statement:** Not applicable.

**Data Availability Statement:** The data presented in this study are available on request from the corresponding author.

**Acknowledgments:** M. Grijalva , R. Parra, A. Rosado, J. Torrico, T. Truong aided in the data collection.

**Conflicts of Interest:** The authors declare no conflict of interest. The funders had no role in the design of the study; in the collection, analyses, or interpretation of data; in the writing of the manuscript, or in the decision to publish the results.

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
