# Peer review of "Assessing the Drivers behind the Structure and Diversity of Fish Assemblages Associated with Rocky Shores in the Galapagos Archipelago"

_jmse, doi:10.3390/jmse9040375_

Round 1

Reviewer 1 Report

The study describes fish species diversity and structure of fish assemblies in different type of habitats in two sites on two separate Galapagos Islands. Some more general comments are presented here first and some more specific following.

General comments

The title starts "Assessing the drivers behind the structure and diversity of fish assemblages..." but in the Conclusion sections it is stated that "Although the sampling effort was distributed in such a way to provide reliable estimates of the underlying environmental drivers of fish assemblage structure and diversity, separating anthropogenic from natural stressors remained difficult."

This problem is explained, at least partly, by a poor design of the study which is essentially unreplicated. That is, one polluted and one pristine site are compared. Even so a number of transects within each site, and replicate observations of each transect are used, this "replication" is rather pseudoreplication with respect to the main drivers that are likely to differ between sites. Pollution and associated factors of nutrients but also oceanographic currents and upwelling, distances to other islands and geological phenomenon. In addition, anthropogenic factors such as boat traffic and fishing (not mentioned in the text though).

It is not particularly clear from the Introduction what was the knowledge gap before this study was conducted and hence what was the aim of this study. A whole bunch of different factors and earlier studies are referred to, from immigration and connectivity to fish fauna of various other geographic regions to local conditions of nutrients, and earlier studies are advocating different factors. In particular, the part between lines 37 and 51 is difficult to follow and the message here is not particularly clear.

In the Introduction (and in the last sentence of the Abstract) it is stated that we can learn from a study like this and draw more generic conclusions and propose guidelines. But is this really possible? Again, the problem with lack of replication make the conclusions of the study only applicable to the two sites and islands considered. (Indeed, the islands are also treated as "fixed" factors in the statistic analyses).

There are also issues about the design, as there might well be dependences among the different transects and observations from the same site depending on how you gathered the data - if all transects of one sites was all done at the same time or if you mixed transects from different places during the same date. It is stated that it was at least one minute between two subsequent observations. This sounds very short, and can you be sure that two observations of the same transect following each other are independent? If there was a predator fish in the first observation is it not then likely that there were less fish prey in the next observation?

What is the qualitative difference of anthropogenic pollution and natural upwelling (line 154). I guess in both cases it is much about nutrients?!

Minor comments

Line 47 and many other places. I find it a bit odd to use the references as if these were names like in this sentence. I would suggest either put the name of the persons in the text (and the reference in the end of the sentence) or rewrite the sentences to something like: "Earlier studies have shown....." and again with the references at the end.

Line 56 - here is a question mark after a reference

Line 63 - "the coastal areas......were assessed using video transects"   The impression here is that this is done from the air and not submarine. Rewrite?

Line 66 - if "multiscale approach" is introduced already here, you need to describe it a bit more. Alternatively, wait with this detail until later where you can also describe what this is.

Line 70- What you mean by intermediate scale is not at all clear. Intermediate to what? Some readers might interpret this as 100 m while other might think of 100 km.....

In next sentence you use "fine scale" and "local scale" what is the difference? Give meters or relevant measures instead at least for fine scale.

Is a place with "approximately 111 inhabitats" not a village rather than a city? And why "approximate" the figures seems rather precise!

Line 105. How were the positions of the lines identified, or where the placing at random?

What was the depth of the transect? What was the time interval between each repeat? Where all the transects in one location done the same day and then the next day another location was done, or did you randomize the order of the transects so that not all transects of one location, say the first, was done in the start of the study, etc.

The line on 167-168 I do not understand.

Lines 174-176, did you analyse your data for any trends from first observation to last observation of the same transect - if the number of scary species declined in a systematic way?

Lines 331-  If a test show statistical significance it means there is a difference and you need not repeat the word "significant" in every sentence, for example, it is enough to say "In Floreana, nitrite and sulfate concentrations were higher, while in ....were higher."

On the other hand, if there is no significant difference you can not (as on line 336) say that values were different "Although not significant, nitrate......were on average higher in Santa Cruz....."  Also note that on line 341 you have a "lack of significant differences" and this cannot be interpreted as if there is no difference- with a non-significant test you simply do not know.

Line 334 - What is "DO concentrations"??

The whole sections 3.1. would be much easier to follow if all the "significances" in the text is removed and only the differences that are statistically supported are reported.

I suggest moving Table 1 into the supplement. Most of this is in the text anyhow.

3.2. Same comment on the use of "significant" in this sections. Furthermore, I do not understand what you mean by "point species richness" and this concept is not explained in the Materials and Methods either.

Table 4. What does "Y" stand for? Explain in Table legend

Line 494. "Tropical"

Reviewer 2 Report

I feel that the paper is interesting and well written. But I have a suggestion for improvement. How about making a diagram for the process of the research? It will be helpful to understand your methodology. And figure 1 needs the source information about where you get the maps. And there are some tables in the result section. How about changing into graphs for easy understanding?

Reviewer 3 Report

I've no remarks, it's solid work made on large material and discussing the essential topic using several methods of modern statistics. I can only subscribe to authors' remark that research should be widen on other localities and larger sample size.   

Reviewer 4 Report

This is an interesting effort to intensively monitor several locations on two of the Galapagos Islands for fish assemblages, and to determine ecological drivers, if possible. The authors were not able to see differences in diversity but did find different species in different conditions. This is not surprising but it is interesting given the location and the difference in human impact on the two sites. My recommendation is to clarify the study design somewhat and to report more about the assemblages themselves, rather than focusing mostly on the ecological conditions.

Lines 118-128, this was hard for me to understand. Revise for clarity: Each location had three transects that were monitored six times by each of three observers on the same day. Five transect locations on each island were monitored within a 12 day period Aug. 19 -31.

(If that is not correct, please explain it so that the reader can understand it without referring to the supplemental material and working it out.)

The supplemental material is useful. The maps should indicate Santa Cruz and Floreana Islands. Fig. S3.2, point out that the different depths have non-overlapping color scales using the same colors.

Table S5.1 shows the list of fishes. Since the purpose of the article is to examine the fish assemblages it would be good to include a summary of this table in the primary text, showing the species that are found in both Islands and those that are unique to each, perhaps including habitat designations where these were seen and number of observations at each location.

Round 2

Reviewer 1 Report

While the authors have clarified many of the unclear messages, I still find the study of limited general value, and indeed the authors themselves still argue that the factor island is a fixed factor, which will then only allow conclusions to be drawn that are applicable to these two specific islands.

This said, I think the study can provide a useful baseline (as also mentioned) for temporal studies for these two specific islands and I am happy to support its publications in JMSE.